# Increased Gene Expression of *C1orf74* Is Associated with Poor Prognosis in Cervical Cancer

**DOI:** 10.3390/cells12212530

**Published:** 2023-10-27

**Authors:** Preetiparna Parida, Shirley Lewis, Krishna Sharan, Mehta Vedant Kamal, Naveena A. N. Kumar, Vishwapriya M. Godkhindi, Sooryanarayana Varambally, Vivek M. Rangnekar, Mahadev Rao, Rama Rao Damerla

**Affiliations:** 1Department of Medical Genetics, Kasturba Medical College, Manipal, Manipal Academy of Higher Education (MAHE), Manipal 576104, Karnataka, India; preetiparna.parida@learner.manipal.edu; 2Department of Radiotherapy and Oncology, Manipal Comprehensive Cancer Care Centre, Kasturba Medical College, Manipal, Manipal Academy of Higher Education (MAHE), Manipal 576104, Karnataka, India; shirley.salins@manipal.edu (S.L.); drkrishna.sharan@nitte.edu.in (K.S.); 3Department of Surgical Oncology, Manipal Comprehensive Cancer Care Centre, Kasturba Medical College, Manipal, Manipal Academy of Higher Education (MAHE), Manipal 576104, Karnataka, India; vedant.mehta1@learner.manipal.edu (M.V.K.); naveenkumar.an@manipal.edu (N.A.N.K.); 4Department of Pathology, Kasturba Medical College, Manipal, Manipal Academy of Higher Education (MAHE), Manipal 576104, Karnataka, India; vishwapriya.mg@manipal.edu; 5Cellular and Molecular Pathology, Department of Pathology, University of Alabama at Birmingham, Birmingham, AL 35294, USA; svarambally@uabmc.edu; 6O’Neal Comprehensive Cancer Center, University of Alabama at Birmingham, Birmingham, AL 35294, USA; 7Department of Radiation Medicine, College of Medicine, University of Kentucky, Lexington, KY 40536, USA; vivek.rangnekar@uky.edu; 8Markey Cancer Center, University of Kentucky, Lexington, KY 40536, USA; 9Department of Pharmacy Practice, Center for Translational Research, Manipal College of Pharmaceutical Sciences, Manipal, Manipal Academy of Higher Education (MAHE), Manipal 576104, Karnataka, India

**Keywords:** *C1orf74*, gene expression, cervical cancer, prognosis, survival

## Abstract

*C1orf74*, also known as URCL4, has been reported to have higher expression and be associated with poor prognosis in lung adenocarcinoma patients, and its role in regulation of the EGFR/AKT/mTORC1 pathway has been recently elucidated. In the current study, we used publicly available data and experimental validation of *C1orf74* gene expression and its association with prognosis in cervical cancer patients. qRT-PCR was performed using RNA from cervical cancer cell lines and twenty-five cervical cancer patients. Data from TNMplot revealed that mRNA expression of the *C1orf74* gene in primary tumor tissues, as well as metastatic tissues from cervical cancer patients, was significantly higher compared to normal cervical tissues. HPV-positive tumors had higher expression of this gene compared to HPV-negative tumors. qPCR analysis also demonstrated higher expression of *C1orf74* in HPV-positive cervical cancer cell lines and most cervical cancer patients. The promoter methylation levels of the *C1orf74* gene in cervical cancer tissues were lower compared to normal cervical tissues (*p* < 0.05). Collectively, our study indicates that higher expression of the *C1orf74* gene caused by hypomethylation of its promoter is associated with poor overall survival in cervical cancer patients. Thus, *C1orf74* is a novel prognostic marker in cervical cancer.

## 1. Introduction

Cervical cancer is the fourth most common cancer in women worldwide [1]. According to Globocan 2022, around 604,000 new cases of the disease are identified each year, and 342,000 deaths were reported in 2020. Persistent infection with high-risk Human papillomavirus (HPV) is the main cause of cervical cancer [1]. However, infection with high-risk HPV is not adequate to cause cervical carcinogenesis. Other genetic events that are either independent of HPV or associated with HPV infection are required for the malignant transformation [2]. Although improvements in screening and preventative methods have decreased the incidence and mortality rates of cervical cancer [3], comprehensive knowledge of the molecular pathways associated with its onset and progression is still required. The majority of cervical cancer patients in India present to hospitals with locally advanced disease. Delay in seeking medical attention is often due to mild or absent symptoms in the early stages, overlooked subtle symptoms such as abnormal vaginal bleeding or pelvic pain—which may be mistaken for other, less serious conditions—limited awareness and education, misdiagnosis or delayed diagnosis, and irregular or no screening of the disease [4,5,6]. 

Cervical cancer patients diagnosed with a localized disease show a better prognosis with a 5-year survival rate of around 91% [7]. However, a huge proportion of patients report to hospitals at advanced stages where treatment options are limited, leading to a 5-year survival rate of less than 17–20% [7,8]. The standard of care treatment for locally advanced cervical cancer often entails a mix of various modalities of therapy. Radiotherapy with chemotherapy is the standard course of treatment for locally advanced cervical cancer [8]. A multidisciplinary team of specialists, including gynecologic oncologists, radiation oncologists, medical oncologists, and other healthcare providers, usually decides on the best course of action for treating locally advanced cervical cancer. The purpose of treatment is to control cancer, improve overall survival, lessen side effects, and maintain patients’ quality of life. Individuals with the same clinical stage and pathological type tend to receive the same treatment, but their prognoses vary, primarily because of their genetic heterogeneity [9]. It is therefore empirical to understand the molecular diversity of the disease and to identify various molecular signatures that may help develop prognostic markers and response-specific molecular signatures of the disease to provide precision-based therapeutic approaches leading to favorable oncological outcomes [10]. Numerous research efforts have been invested in identifying prognostic markers that could be used to enhance patient care and survival [11]. Among these markers, HPV status (especially HPV-16 and HPV-18), p16^ink4a^ overexpression, *TP53* mutations, and mTOR pathway activation are a few pertinent molecular signatures of cervical cancer that are associated with an aggressive disease with a poor prognosis [12]. With the advent of high throughput genomic technologies like DNA microarray, it has been possible for researchers to identify such prognostic markers [13]. Several studies have reported gene expression patterns in cervical cancers that can predict recurrence. Huang et al. identified an 11-gene signature in cervical cancer that can predict pelvic lymph node metastasis in cervical carcinoma and may guide clinicians in planning the therapy for the patient [14]. They have also identified a 7-gene (*UBL3, FGF3, BMI1, PDGFRA, PTPRF, RFC4*, and *NOL7*) signature predicting relapse in patients with cervical cancer [11].

Recently, there has been a significant improvement in research efforts focusing on exploring expression patterns of several genes among cohorts of patients with cervical cancer. These studies have delineated the crucial role of gene expression analyses in understanding the critical landscape of cervical cancer [15,16,17,18]. Lee et al. used an NGS (Next Generation Sequencing) panel consisting of major cancer driver and tumor suppressor genes and demonstrated that mutations in the *RNF213* gene could be used as a treatment monitoring marker [19]. Similarly, a recent study by Liu et al. found the association of *KMT2C* and *LRP1B* gene mutations with high tumor mutational burden in cervical cancer patients. They also reported that mutations in *PTEN* were associated with poorer survival in patients with cervical cancer [20]. Bioinformatics tools to visualize data from omics projects such as cBioportal and UALCAN have emerged as essential components for these analyses, as they include comprehensive information regarding gene expression patterns, potential biomarkers, as well as clinical outcomes from various studies performed worldwide [21,22,23]. UALCAN is one of these useful resources. This platform provides researchers with a plethora of knowledge on expression profiles, survival profiles, and other clinical outcomes. It also provides a list of top genes whose expressions are closely associated with prognosis in cervical cancer patients [21,22,23,24]. Among the list of these top identified genes through integrative analysis on UALCAN, *C1orf74* emerged as a promising candidate. *C1orf74* is a protein-coding gene present on chromosome 1 at locus 1q32.2 in humans between the genes *TRAF31P3* and *IRF6* [25]. It has two exons, out of which only one is coding [26]. The transcript of *C1orf74* results in an mRNA that is 4684bp and codes for the UPF0739 protein C1orf74 with 269 amino acids and a molecular weight of 29,561 Daltons (Da) [27]. The neighboring genes of *C1orf74* are *TRAF3IP3* and *IRF6* [28]. Fusion transcripts of *C1orf74* and neighboring *IRF6* have been reported previously in squamous cell carcinomas [29]. The C1orf74 protein is found in both cytoplasm as well as plasma membrane but its function in normal and cancer cells remains unknown [25]. A recent research study reported that higher expression of *C1orf74* was associated with poorer prognosis in lung adenocarcinoma patients. It was also observed that *C1orf74* can positively regulate the EGFR/AKT/mTORC1 pathway, affecting cell proliferation and mobility [25]. It is well known that cervical cancer develops and spreads as a result of the EGFR/AKT/mTORC1 signaling pathway and its connection to HPV infection. The interaction of cervical cancer’s EGFR/AKT/mTORC1 signaling pathway with HPV infection leading to cervical cancers forms a strong plinth to determine the role of *C1orf74* in oncogenesis and further progression of the disease. We evaluated the expression of the *C1orf74* gene in patients with cervical cancer and assessed its association with various pathways to further correlate the expression patterns with clinical outcomes with further progression of the disease. Our study suggests that *C1orf74* is a valuable prognostic marker for cervical cancer.

## 2. Materials and Methods

### 2.1. Expressions of Genes Associated with Survival in Cervical Cancer Patients

Transcription levels of various genes associated with survival in cervical cancer patients were analyzed using UALCAN. This resulted in a list of top genes whose gene expression is correlated to overall survival in cervical cancer data from TCGA (The Cancer Genome Atlas) RNAseq studies [21,22,23]. Kaplan–Meier survival plots were generated for each gene in each TCGA cancer type and the log-rank test was used to compare the survival curves of samples with high gene expression and samples with low/medium gene expression [22].

### 2.2. Expression Level of the C1orf74 Gene in Different Types of Cancers 

Transcription levels of *C1orf74* in pan-cancers were obtained using Gene Expression Profiling Interactive Analysis (GEPIA) (http://gepia.cancer-pku.cn/) (accessed on 17 July 2023) [30,31]. GEPIA comprises information from the TCGA and GTEx databases on 9736 tumors and 8587 normal tissues. The expression of the *C1orf74* gene in various cancers was also examined using UALCAN (http://ualcan.path.uab.edu) (accessed on 17 July 2023) [21,22,23]. The TNMplot database was used to retrieve TCGA RNA Seq data to explore the expression of the *C1orf74* gene in cervical cancer tumors and normal and metastatic tissue (https://tnmplot.com/analysis/) (accessed on 10 July 2023) [32,33]. The UALCAN database was used to obtain expression levels of *C1orf74* across different cancer stages and histological types of cervical cancer (http://ualcan.path.uab.edu) (accessed on 17 July 2023) [21,22,23]. We also compared the expression of *C1orf74* in HPV-positive and HPV-negative cervical cancer cases using the OncoDB portal (https://oncodb.org/) (accessed on 17 July 2023) [34,35]. *C1orf74* expression levels at different pathological stages were also analyzed using OncoDB [34,35], which uses the clinical data from TCGA. 

### 2.3. Patient Samples 

Twenty-five different tumor and blood pairs from patients with various stages of cervical cancer were evaluated for comparison of transcript levels of *C1orf74.* All patients were recruited at Kasturba Medical College, Manipal, and paired samples (tumor and blood) were collected after obtaining informed consent. Ethical approval was obtained from the Institutional Ethics Committee, Kasturba Medical College Manipal, Manipal Academy of Higher Education, Manipal, India. All the tumor samples were characterized by a certified pathologist. The clinical parameters like age, histological grade, comorbidity status, etc., were obtained from the medical records and summarized in Table 1. Tumor samples were snap-frozen using liquid nitrogen or dry ice ethanol bath and stored at −80 °C until further use. Primary fibroblast cells derived from tissues obtained from skin biopsies from healthy individuals were used as controls. 

### 2.4. Cell Culture 

Cervical cancer cell lines SiHa, Ca Ski, (HPV-16 positive), HeLa (HPV-18 positive), and C33A (HPV negative) were maintained in Dulbecco’s modified Eagle’s medium (DMEM) (HiMedia Laboratories Pvt. Ltd., Mumbai, India) with 10% fetal bovine serum (HiMedia Laboratories Pvt. Ltd., Mumbai, India) and 1% antibiotic/antimycotic solution. The cell lines were procured from the National Centre for Cell Sciences (NCCS), Pune, India. Fibroblasts were grown in DMEM supplemented with 10% FBS and 1% antibiotic/antimycotic solution. Cells were maintained at 37 °C in an atmosphere of 5% CO_2_.

### 2.5. DNA Isolation, HPV Detection and Genotyping

DNA was extracted from tumor samples using a DNeasy Blood and Tissue kit (Qiagen, Hilden, Germany) following the manufacturer’s protocol. HPV detection by PCR was performed using primers GP5+ (TTTGTTACTGTGGTAGATACTAC) and GP6+ (GAAAAATAAACTGTAAATCATATTC). Further subtyping of HPV was determined using E6 primers: HPV16E6F-TATGCACAGAGCTGCAAACA and HPV16E6R-GCAAAGTCATATACCTCACGTC for HPV16; HPV18E6F-ATGGCGCGCTTTGA and HPV18E6R-CTGTAAGTTCCAATACTGTCTTG for HPV18; and HPV31E6F-GAAATTGCATGAACTAAGCTCG and HPV31E6R-CACATATACCTTTGTTTGTCAA for HPV31. 

### 2.6. RNA Isolation, Reverse Transcription, and Real-Time Quantitative Polymerase Chain Reaction (RT-qPCR)

Total RNA was extracted from cell lines using TRIzol™ Reagent (Thermo Fisher, Carlsbad, CA, USA) according to the manufacturer’s protocol. RNA was extracted from blood using an Ambion™ RiboPure™ RNA Purification Kit, blood (Thermo Fisher, Carlsbad, CA, USA). An AllPrep DNA/RNA Mini kit (Qiagen, Hilden, Germany) was used for the extraction of total RNA from tissue biopsy samples. Total RNA was converted into cDNA using an iScript™ gDNA Clear cDNA Synthesis Kit (BioRad, Hercules, CA, USA) according to the manufacturer’s instructions. Primers for amplifying *C1orf74* were designed using Primer3 online software (version 4.1.0) (https://primer3.ut.ee/) (accessed on 17 July 2023). The primer sequences for PCR amplification of *c1orf74* and *GAPDH* from cDNA are as follows: *C1orf74*F-TACAGCAGGCTCCATTCCT, *C1orf74*R-ATGAGATCCGGGCAGTGAAT, GAPDHF-CGACCACTTTGTCAAGCTCA, and GAPDHR-GAGGGTCTCTCTCTTCCTCT. RT-qPCR analysis was performed using TB Green Premix Ex Taq II (Tli RNase H Plus) (Takara Bio Inc., Tokyo, Japan; RR820A) on QuantStudio^TM^ 5 (Applied Biosystems, Waltham, MA, USA). The PCR cycling conditions were as follows: 95 °C for 1 min followed by 40 cycles of 95 °C for 5 s, 50 °C for s, and 72 °C for s. The 2^−ΔΔCt^ method was used for quantification and fold change for the target gene. The values were first normalized to internal reference gene GAPDH, followed by calculating relative expression to healthy control. Statistical analysis was performed using a *t*-test, and a *p*-value < 0.05 was considered statistically significant. 

### 2.7. Promoter Methylation Analysis of C1orf74

The promoter methylation levels of *C1orf74* in cervical cancer were identified using the UALCAN database, which offers gene expression analysis on over 20,500 protein-coding genes in 33 different tumor types using data from the TCGA project. The database also provided the promoter methylation levels of *C1orf74* across the different stages of cervical cancer (http://ualcan.path.uab.edu) (accessed on 17 July 2023) [21,22,23].

### 2.8. Survival Analysis 

We used GEPIA for analyzing the correlation between *C1orf74* expression and survival, including overall survival (OS) and disease-free survival (DFS) (http://gepia.cancer-pku.cn/) (accessed on 17 July 2023) [30,31]. The plot included Cox P or P and hazard risk (HR) values from a log-rank test. The OS was analyzed for HPV-positive and negative cervical cancer data available on the OncoDB portal (https://oncodb.org/) (accessed on 17 July 2023) [34,35]. 

### 2.9. Interaction Network and Functional Enrichment Analyses of C1orf74-Correlated Genes 

To explore the role of *C1orf74* in cervical cancer, expression of genes that positively correlated with *C1orf74* expression in cervical cancer were extracted from UALCAN (http://ualcan.path.uab.edu) (accessed on 17 July 2023) [21,22,23]. The top 25 genes significantly correlated with *C1orf74* expression were chosen for further analysis. Gene ontology (GO) and Kyoto Encyclopedia of Genes and Genomes (KEGG) pathway enrichment analyses were performed and graphically visualized using ShinyGO (version 0.77) (http://bioinformatics.sdstate.edu/go/) (accessed on 10 August 2023) [36]. The significance threshold for the enrichment in ShinyGO was set at the false discovery rate ≤0.05.

## 3. Results

### 3.1. Expression of Genes Associated with Survival in Cervical Cancer

We used TCGA data represented in UALCAN to determine the differential regulation of genes associated with poor overall survival in cervical cancer. The list of the top 10 genes obtained from this analysis is shown in Table 2 [21,22,23]. We found that *C1orf74* was among the top 3 hits and has not been previously studied. 

### 3.2. Expression of C1orf74 in Cervical Cancer 

The gene expression profiles of *C1orf74* across tumor and normal tissues were investigated using the GEPIA database to check the differential expression of *C1orf74* among different cancers [30,31]. The expression of *C1orf74* was higher in most cancers compared to normal tissues. *C1orf74* was especially found to be significantly overexpressed in cervical cancers (CESCs), lung squamous cell carcinomas (LUSCs), and thymoma (THYM) (Figure 1A). RNA-Seq data from the TNMplot database revealed higher expression of the *C1orf74* gene in cervical tumors and metastatic cervical cancer samples compared to normal tissue (Figure 1B) (Appendix A). However, we did not observe any significant difference in the expression of *C1orf74* based on individual cervical cancer stages (Figure 1C) (Appendix A). Among the histological subtypes, squamous cell carcinomas showed the highest levels of expression compared to normal, adenosquamous, endocervical, endometroid, and mucinous tumors. The expression levels of *C1orf74* in all histological subtypes of cervical cancers were significantly higher than the normal control tissues (Figure 1D) (Appendix A). HPV positivity also correlated with higher expression of *C1orf74*. We found that *C1orf74* expression in HPV-positive cervical cancers was significantly higher compared to HPV-negative cervical cancers (Figure 1E) (Appendix A). Whereas the results from OncoDB also depict that *C1orf74* expression is higher based on tumor type (T), the extent of spread to the lymph nodes (N), and the presence of metastasis (M) stages, but the *C1orf74* expression did not show a significant difference between different stages (Appendix A).

### 3.3. C1orf74 Expression in Cervical Cancer Cell Lines and Patient Tumor Samples 

To validate in silico findings from different databases, the mRNA expression of *C1orf74* in four cervical cancer cell lines (SiHa, Ca Ski, HeLa, and C33A) was measured. The selection of these specific cell lines was based on the need to compare the relative expression of *C1orf74* mRNA in HPV-positive cervical cancer cell lines HeLa, SiHa, and Ca Ski versus the HPV-negative cervical cancer cell line C33A. Moreover, the relative gene expression across different HPV subtypes (HPV 16 and HPV18) was also elucidated by comparing HPV 16-positive cell lines (SiHa and Ca Ski) versus the HPV 18-positive cell line (HeLa) (Appendix A). The results suggested that the expression of *C1orf74* was higher in HPV-positive cervical cancer cell lines compared to normal cells and the HPV-negative cancer cell line C-33A derived from a uterine tumor. These results were consistent with data we obtained from OncoDB (Figure 2A), which suggested that cervical tumors have higher expression of *C1orf74* compared to normal tissues. Although there was a notable increase in *C1orf74* gene activity in the HPV 16-positive Ca Ski cell line, the variations in *C1orf74* activity levels in HeLa (HPV 18) and SiHa (HPV 16) cells were not significant. We then evaluated the levels of *C1orf74* expression in tumors derived from 25 cervical cancer patients using RT-qPCR. We observed a significant increase in the expression of *C1orf74* in most of the patients (20 out of 25 patients). And this increase was consistent across all stages of cervical cancer (Figure 2B).

### 3.4. Promoter Methylation Analysis of C1orf74 in Cervical Cancer 

Higher expression of *C1orf74* in tumors prompted us to hypothesize that the promoter of this gene could also be hypomethylated. Promoter DNA methylation levels of *C1orf74* in the cervical cancer TCGA dataset were acquired using UALCAN [21,22,23]. We observed a significant reduction in promoter methylation levels of *C1orf74* in cervical cancer tissues compared to normal tissues, as shown in Figure 3A (Appendix A). Further analysis found that the promoter methylation level of *C1orf74* was consistently lower in all stages of cervical cancer compared to normal tissues (Figure 3B) (Appendix A). These data indicate that the expression of *C1orf74* may be related to the degree of methylation in the promoter region. We also analyzed methylation levels in various tumor histologies of cervical cancer. The promoter methylation levels of *C1orf74* in squamous cell carcinomas were found to be the lowest, followed by adenocarcinomas and endocervical carcinomas Appendix A. 

### 3.5. Survival Analysis 

The association of *C1orf74* expression with the overall survival of cervical cancer patients was analyzed using the GEPIA server (http://gepia.cancer-pku.cn/) (accessed on 17 July 2023) [30]. The overall survival of patients with higher expression of *C1orf74* was associated with significantly poor outcomes compared to patients with low *C1orf74* expression (*p* = 0.0019) (Figure 4A). We also analyzed the levels of *C1orf74* gene expression in HPV-positive cervical cancer patients using OncoDB (https://oncodb.org/) (accessed on 17 July 2023) (Figure 4B) [34,35]. Moreover, we observed that HPV-positive patients with higher expression of *C1orf74* had poorer survival compared to HPV-positive patients with lower expression.

### 3.6. Interaction Network, Prognostic Impact, and Functional Enrichment of C1orf74-Correlated Genes in Cervical Cancer 

In this study, we obtained the top 25 genes (Appendix A) with the highest correlation with *C1orf74* in cervical cancer from UALCAN for further analysis to define the possible role of *C1orf74* in the development of cervical cancer [21,22,23]. We used this list of correlated genes for functional enrichment analysis using ShinyGO. The GO molecular function data showed the enrichment of genes from the pathways of Hippo signaling, transcriptional misregulation in cancer, viral carcinogenesis, and Ras signaling (Figure 5A). The KEGG function suggested the role of *C1orf74* in several important signaling pathways, including those regulated by the tumor suppressor p53, the cervical cancer vulnerability hormone estrogen, oncogenic Ras, proliferation-associated MAPK, and cell survival PI3K-Akt signaling (Figure 5B).

## 4. Discussion

Cervical cancer is curable in its early stages [37], but patients usually present themselves at advanced stages of the disease, especially in low- and middle-income countries, where cervical cancer remains a major cause of mortality in women [9,38]. Therefore, it is vital to explore suitable biomarkers for prognosis and prediction of metastasis in patients with cervical cancer. We analyzed the TCGA dataset available on UALCAN for querying genes that affect the survival and prognosis of cervical cancer (Table 2). *C1orf74* emerged as a promising biomarker during this analysis, with higher expression of *C1orf74* being associated with low overall survival. An extensive literature search identified only one publication suggesting a role for *C1orf74* in regulating EGFR/AKT/mTORC1 signaling lung adenocarcinoma cells [25].

We observed significantly higher expression of *C1orf74* in HPV-positive cervical cancer cell lines HeLa, SiHa, and Ca Ski. Although all three of these HPV-positive cell lines showed significantly higher expression of *C1orf74*, we observed that Ca Ski showed markedly higher expression of *C1orf74* than Hela and SiHa. It is interesting to note that HeLa and SiHa cell lines have fewer than two copies of HPV integrated into the human genome, while Ca Ski has multiple copies of HPV in an integrated form. Analysis of mRNA expression profiles of *C1orf74* in cervical cancer TCGA datasets using multiple online resources like UALCAN, GEPIA, TNMplot, and OncoDB revealed that cervical tumors have significantly higher expression levels compared to other cancers. We also observed that HPV-positive cervical cancers showed higher expression of *C1orf74* than the HPV-negative group of cervical cancers. The TCGA gene expression data correlated with qPCR analysis on mRNA derived from 25 HPV-positive patients from our cohort. Our cohort included three metastatic patients, but only one of them showed a higher expression than the normal control. It however remains unclear whether *C1orf74* plays any role in metastasis in cervical cancer. We could not see any significant difference in the expression level of *C1orf74* across the pathological T, N, and M stages (Appendix A). Moreover, *C1orf74* was not found in the list of the top 200 genes whose expression levels are significantly different (Appendix A) in these pathological stages of cervical cancer in OncoDB. But the analysis of *C1orf74* expression in HPV-positive patients from OncoDB revealed a higher expression of *C1orf74* correlated well with overall worse survival compared to HPV-positive patients with lower expression of the gene. Additionally, there are huge deviations in expression levels of *C1orf74* from *in silico* analysis and also from our patient cohort. This is similar to what we observed in HPV-positive cell lines, where the number of copies of HPV and the location of HPV integration might influence *C1orf74* expression. It would be interesting to correlate *C1orf74* expression with the HPV integration state and loci in these patients, as observed in the aforementioned HPV-positive cell lines.

Epigenetic regulation of gene expression through DNA methylation and post-translational modifications of histones at specific loci can result in uncontrolled cell proliferation and cervical carcinogenesis [39]. CpG islands are usually present at 5’ promoter regions of many genes [40]. Under normal physiological conditions, the promoter regions are methylated in oncogenes such as *MYC* and *HRAS*, and hypomethylation of promoter regions of oncogenes can cause overactivation of genes and contribute to oncogenesis [41]. An increase in the expression levels of oncogenes can be correlated with hypomethylation of gene promoters. Consistently, we found that the methylation levels of the *C1orf74* promoter were lower in cervical cancer tissues than the normal tissues, and this reduced methylation was maintained across all stages of cervical cancer. Further functional analysis of *C1orf74* will help understand if the differences in promoter methylation levels indeed contribute to disease progression. 

Enrichment analysis in GO indicated that the *C1orf74*-associated genes were closely associated with various cellular pathways involving Hippo signaling, viral carcinogenesis, and transcription (Figure 5A). These data further corroborate the strong correlation between *C1orf74* expression and its potential function in HPV-associated carcinogenesis of cervical tumors. KEGG pathway analysis revealed that the co-expressed genes were enriched in multiple pathways including p53, Ras, MAPK, and PI3K-Akt pathways (Figure 5B). These molecular pathways have previously been reported to play crucial roles in cervical carcinogenesis [42]. This enrichment of the pattern of the PI3K-Akt pathway positively correlated with *C1orf74* expression in cervical cancer. Our findings are in agreement with the previous report of *C1orf74* in lung adenocarcinoma, which is not hormonally regulated like cervical cancer, but showed a correlation of *C1orf74* with the regulation of the EGFR/AKT/mTORC1 pathway [25]. The PI3K/Akt signaling pathway plays a crucial role in the Ras-mediated transformation of cancer cells [43]. As the PI3K pathway is often involved in the regulation of cell adhesion molecules such as E-Cadherin and β-Catenin, it plays a key role in the attenuation of cell adhesion and promotion of motility of cancer cells, which is the major cause of invasion and metastasis [44]. Our KEGG pathway enrichment analysis also indicated that the co-expressed genes were significantly enriched in the focal adhesion process. Although the expression of *C1orf74* was not significantly different among various stages of cervical tumors, our findings suggest that *C1orf74* may be involved in the invasion and metastasis of cervical cancer.

Recent studies have shown the EGFR/AKT/mTORC1 signaling pathway plays a crucial role in HPV-induced cancers [45]. It has been also reported that the HPV oncogenes E6/E7/E5 usually activate this pathway for modulation of tumor growth and progression, and hence can play a major role in patient survival and clinical outcomes [46,47,48]. The association of EGFR/AKT/mTORC1 pathways with *C1orf74* [25] and the crucial role of HPV in this pathway identified by our study justifies *C1orf74* as an ideal candidate marker for predicting survival. 

## 5. Conclusions

In the current study, we analysed the expression and prognostic value of *C1orf74* in cervical cancer. These results suggested that *C1orf74* overexpression correlated with HPV positivity and a significantly lower survival and prognostic value in patients with cervical cancer. We also validated these findings in our cohort of cervical cancer patients using qRT-PCR. Our comprehensive analysis revealed several enriched pathways such as those regulated by oncogenic Ras, PI3K-AKT, viral carcinogenesis, and transcriptional deregulation that were significantly associated with elevated expression of the *C1orf74* gene. Combining this information with our understanding of *C1orf74* protein function, we infer that *C1orf74* plays a role in promoting the development of cervical cancer. Thus, *C1orf74* presents a new prognostic marker and novel target for cervical cancer treatment. Our findings can be extended to studies on the molecular mechanisms of *C1orf74* in diverse cancers. To our knowledge, this is the first study reporting the role of *C1orf74* in cervical cancer. 

## Figures and Tables

**Figure 1 cells-12-02530-f001:**
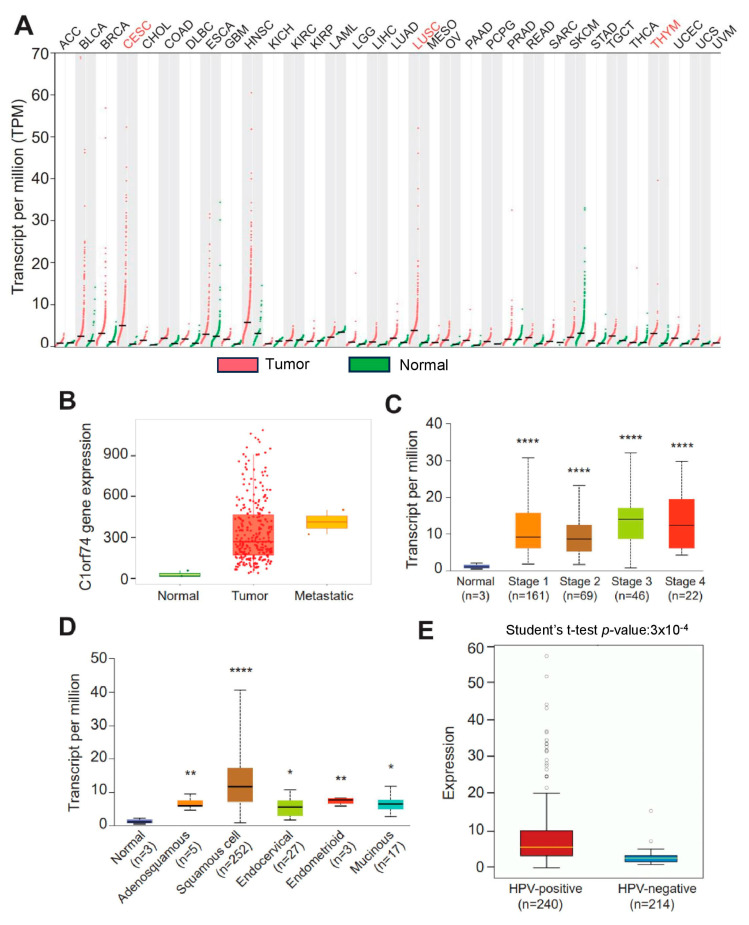
*C1orf74* expression levels in different human tumors and in cervical cancers. (**A**) GEPIA database-based expression of *C1orf74* in different tumors compared to normal tissue. ACC:T(n = 77) N(n = 128), BLCA1:T(n = 403) N(n = 28), BRCA:T(n = 1085) N(n = 291), CESC:T(n = 306) N(n = 13), CHOL:T(n = 36) N(n = 9), COAD:T(n = 275) N(n = 349), DLBC:T(n = 47) N(n = 337), ESCA:T(n = 182) N(n = 286), GBM:T(n = 163) N(n = 207), HNSC:T(n = 519) N(n = 44), KICH:T(n = 66) N(n = 53), KIRC:T(n = 523) N(n = 100), KIRP:T(n = 286) N(n = 60), LAML:T(n = 173) N(n = 70), LGG:T(n = 518) N(n = 207), LIHC:T(n = 369) N(n = 160), LUAD:T(n = 483) N(n = 347), LUSC:T(n = 486) N(n = 338), MESO:T(n = 87), OV:T(n = 426) N(n = 88), PAAD:T(n = 179) N(n = 171), PCPG:T(n = 182) N(n = 3), PRAD:T(n = 492) N(n = 152), READ:T(n = 92) N(n = 318), SARC:T(n = 262) N(n = 2), SKCM:T(n = 461) N(n = 558), TAD:T(n = 408) N(n = 211), TGCT:T(n = 137) N(n = 165), THCA:T(n = 512) N(n = 337), THYM:T(n = 118) N(n = 339), UCEC:T(n = 174) N(n = 91), UCS:T(n = 57) N(n = 78), UVM:T(n = 79). (**B**) TNMplot data for the expression of *C1orf74* gene in cervical cancer tumors (n = 304), normal (n = 3), and metastatic tissue (n = 2) (*p*-value 1.04 × 10^−2^ Kruskal–Wallis test). (**C**) UALCAN analysis of expression profiles of *C1orf74* across different stages of cancer. (**D**) UALCAN analysis of expression profiles of database provided the expression levels of *C1orf74* across different histological types of cervical cancer (http://ualcan.path.uab.edu) (accessed on 17 July 2023). (**E**) Comparison of mRNA expression of *C1orf74* in HPV-positive and HPV-negative cervical cancers (OncoDB) (https://oncodb.org/) (accessed on 17 July 2023). Abbreviations: ACC: adrenocortical carcinoma; BLCA: bladder urothelial; BRCA: breast invasive carcinoma; CESC: cervical squamous cell carcinoma; CHOL: cholangiocarcinoma; COAD: colon adenocarcinoma; DLBC: lymphoid neoplasm diffuse large B cell lymphoma; ESCA: esophageal carcinoma; GBM: glioblastoma multiforme; HNSC: head and neck squamous cell carcinoma; KICH: kidney chromophobe; KIRC: kidney renal clear cell carcinoma; KIRP: kidney renal papillary cell carcinoma; LAML: acute myeloid leukemia; LGG: brain lower grade glioma; LIHC: liver hepatocellular carcinoma; LUAD: lung adenocarcinoma; LUSC: lung squamous cell carcinoma; MESO: mesothelioma; OV: ovarian serous cystadenocarcinoma; PAAD: pancreatic adenocarcinoma; PCPG: pheochromocytoma and paraganglioma; PRAD: prostate adenocarcinoma; READ: rectum adenocarcinoma; SARC: sarcoma; SKCM: skin cutaneous melanoma; STAD: stomach adenocarcinoma; TGCT: testicular germ cell tumors; THCA: thyroid carcinoma; THYM: thymoma; UCEC: uterine corpus endometrial carcinoma; UCS: uterine carcinosarcoma; and UVM: uveal melanoma; (* *p* < 0.05, ** *p* < 0.01, **** *p* < 0.0001, ns = not significant). *p*-values are mentioned in Appendix A.

**Figure 2 cells-12-02530-f002:**
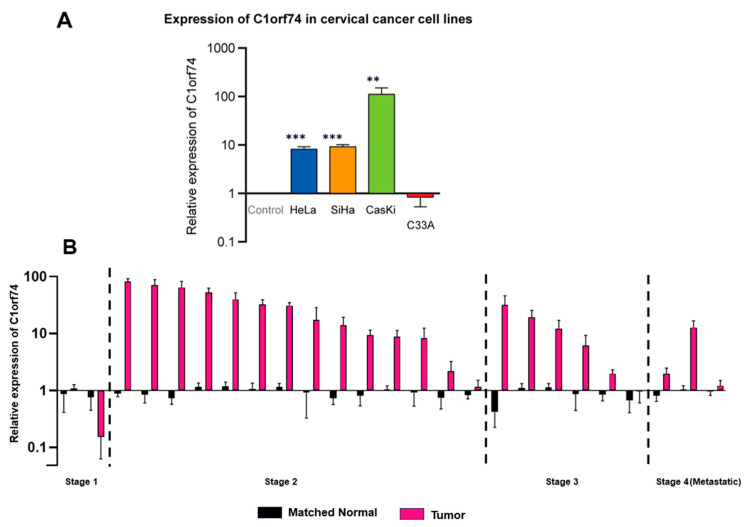
(**A**) Relative expression of *C1orf74* in cervical cancer cell lines SiHa, HeLa, Ca Ski, and C33A. ** *p* < 0.01, *** *p* < 0.001, ns = not significant. *p*-values are mentioned in Appendix A. (**B**) Relative mRNA expression of *C1orf74* in 25 cervical patients compared with healthy control fibroblast cells. Paired *t*-tests indicated differences in log expressions between “Tumor” and paired normal (*p* = 0.0003).

**Figure 3 cells-12-02530-f003:**
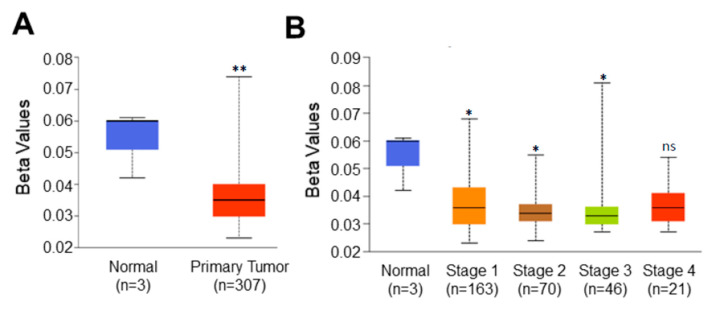
Promoter methylation level of *C1orf74* in cervical cancer. (**A**) Comparison of *C1orf74* promoter methylation levels in normal and cervical cancer tissues (UALCAN). (**B**) Promoter methylation levels of *C1orf74* gene across different stages of cervical cancer; (* *p* < 0.05, ** *p* < 0.01, ns = not significant). *p*-values are mentioned in Appendix A. (The beta value indicates level of DNA methylation ranging from 0 (unmethylated) to 1 (fully methylated)) [21].

**Figure 4 cells-12-02530-f004:**
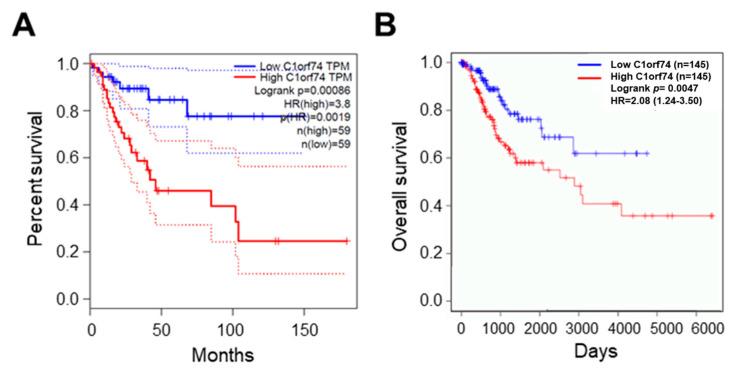
Survival analysis of cervical cancer patients with high and low expression of *C1orf74.* (**A**) OS curve for expression of *C1orf74* in cervical cancer from GEPIA. (**B**) OS curve for expression of *C1orf74* in HPV-positive and HPV-negative cervical cancers from OncoDB. OS = overall survival. Solid lines represent median and dotted lines represent 95% confidence intervals.

**Figure 5 cells-12-02530-f005:**
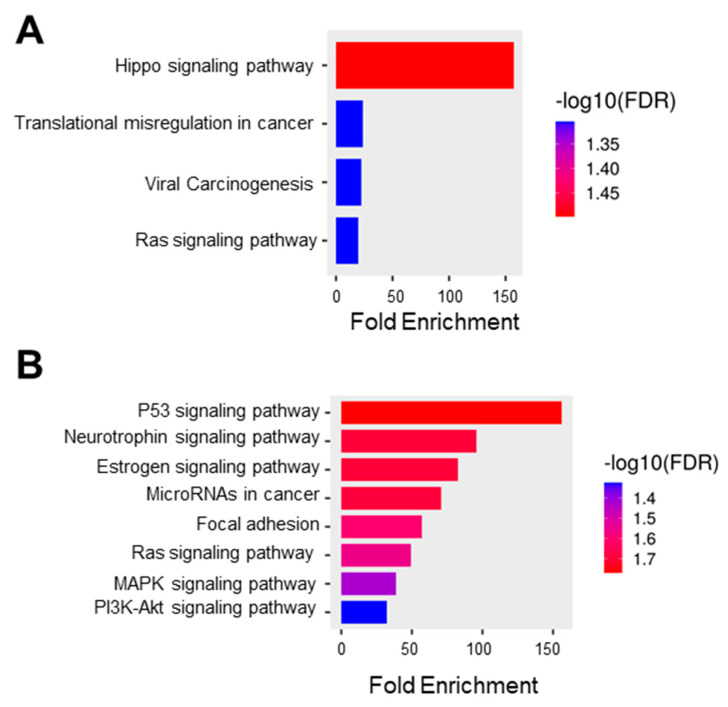
Functional protein association network and pathway analysis. (**A**) GO molecular function database (ShinyGO) (**B**) KEGG pathway enrichment where the bar plot illustrates the enriched pathways (FDR < 0.05) (ShinyGO).

**Table 1 cells-12-02530-t001:** Demographics and clinical characteristics of patients.

Patients		n = 25
Age		
	Median	52 years
	Range	38–73 years
Race		Indian
Ethnicity		South Indian
Histology		
	Squamous cell carcinoma	24
	Adenocarcinoma	1
Grade		
Squamous cell carcinoma		
	Grade 1	1
	Grade 2	9
	Grade 3	6
	Not determined	8
Adenocarcinoma		
	Grade 2	1
Stage		
	1	3
	2	13
	3	6
	4 (metastatic)	3
HPV		
	Positive	25
	Negative	0
HPV type		
	HPV16	22 *
	HPV18	2
	HPV 31	1
HIV		0
HBV		0
HCV		0
Presenting complaints		
	White discharge per vaginum	13
	Pain (abdominal/suprapubic)	13
	Post-menopausal bleeding	11
	Bleeding per vaginum	4
	Foul-smelling discharge per vaginum	3
	Postcoital bleeding	3
	Urinary symptoms	2
	Hematuria	1
	Loose motions	1
	Pyometra	1
	Spotting	1
Comorbidities		
	Hypertension	5
	Diabetes mellitus	3
	Obesity	2
	Anemia	2
	Hydroureteronephrosis	2
	Depression	1
	Hypothyroidism	1
	Acute kidney injury	1
	Total knee replacement	1
	L4-L5 intervertebral disc prolapse	1
	Nil comorbidities	12

* Single adenocarcinoma was positive for HPV16.

**Table 2 cells-12-02530-t002:** Top genes based on Kaplan–Meier survival analysis in CESC in UALCAN webportal.

SL No.	Gene	Log Rank *p*-Value	High Expression: Median Survival (in Days)	Low Expression: Median Survival (in Days)
1	*EREG*	2.80 × 10^−6^	1245	NA
2	*SLN*	3.70 × 10^−6^	1186	NA
3	*C1orf74*	4.20 × 10^−6^	1372	NA
4	*PCDHB17*	4.30 × 10^−6^	955	4086
5	*SCAND3*	9.80 × 10^−6^	1186	4086
6	*ANKRD34B*	1.06 × 10^−5^	NA	2520
7	*ESM1*	1.25 × 10^−5^	1011	NA
8	*PAPPA*	2.04 × 10^−5^	1065	NA
9	*LOC90586*	5.06 × 10^−5^	1210	NA
10	*ZNF134*	5.29 × 10^−5^	1186	NA

NA, not available.

## Data Availability

The data used in this article are freely available in The Cancer Genome Atlas (TCGA) database (https://portal.gdc.cancer.gov) (accessed on 17 July 2023), cBioportal (https://www.cbioportal.org/) (accessed on 17 July 2023), UALCAN (http://ualcan.path.uab.edu) (accessed on 17 July 2023) and ShineyGO (http://bioinformatics.sdstate.edu/go/) database (accessed on 17 July 2023), GEPIA (http://gepia.cancer-pku.cn/) (accessed on 17 July 2023), and CCLE (www.broadinstitute.org/ccle) (accessed on 17 July 2023), respectively. OncoDB (https://oncodb.org/) (accessed on 17 July 2023). The qPCR data used to support the findings of this study are available within the article and from the corresponding author upon request.

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
