# Peer review of "Increased Gene Expression of C1orf74 Is Associated with Poor Prognosis in Cervical Cancer"

_cells, 2023, doi:10.3390/cells12212530_

Round 1
Reviewer 1 Report
Comments and Suggestions for Authors
The authors use the results from databases such as TCGA, UALCAN, and oncoDB to demonstrate that increased gene expression of C1orf74 is associated with HPV infection and poor prognosis. Their results and discussion were almost convincing. I have some minor comments.
1. Could you provide more information about C1orf74, such as mRNA length, protein MW, containing domains, the nearby genes in the genome, and promotor analysis?
2. Figure 5 was missing. In line 286, figure 5B should be 4B, and no cited Figure 4A.
3. Figure 1A's X-axis labels were too small to see.
4. Are the "stage I, II, III, IV" in Figure 2B equal to "stage 1, 2, 3, 4" in other Figures? If they are the same, please use consistent labels.
5. Please define the T, N, and M stages of line 212.
6. What is the "Beta Values" in Figure 3?
7. In line 211, Oncodb should be changed to OncoDB for consistent usage.
Author Response
Author's Reply to the Review Report (Reviewer 1)
Suggestions for Authors
The authors use the results from databases such as TCGA, UALCAN, and OncoDB to demonstrate that increased gene expression of C1orf74 is associated with HPV infection and poor prognosis. Their results and discussion were almost convincing. I have some minor comments.
- Could you provide more information about C1orf74, such as mRNA length, protein MW, containing domains, the nearby genes in the genome, and promotor analysis?
We added additional details on C1orf74 - mRNA length, protein molecular weight, neighbouring genes in paragraph 3 of the Introduction.
- Figure 5 was missing. In line 286, figure 5B should be 4B, and no cited Figure 4A.
We have corrected the error by changing Figure 6 to Figure 5. Figure 5B was corrected to 4B (Line 339) and Figure 4A is cited in line 337.
- Figure 1A's X-axis labels were too small to see.
We uploaded a high-resolution picture along with changing the x-axis labels and also included detailed information regarding the number of samples (n) in the figure legend.
- Are the "stage I, II, III, IV" in Figure 2B equal to "stage 1, 2, 3, 4" in other Figures? If they are the same, please use consistent labels.
Thank you for bringing this to our attention. We made the correction and have now used stage 1, 2, 3 and 4 consistently throughout the manuscript.
- Please define the T, N, and M stages of line 212.
We have defined the T, N and M stages in lines 252-253 as Tumor (T), extent of spread to the lymph nodes (N), and presence of metastasis (M).
- What is the "Beta Values" in Figure 3?
The “Beta Values” indicate level of DNA methylation ranging from 0 (unmethylated) to 1 (fully methylated). This is now clarified in Figure 3 legend.
- In line 211, Oncodb should be changed to OncoDB for consistent usage.
As you suggested, we changed Oncodb to OncoDB in line 251.
Reviewer 2 Report
Comments and Suggestions for Authors
The subject of the manuscript is actual and relevant, and this is a good piece of work. The paper is clearly written providing the necessary information (some shortcomings, see below). The observed results are discussed and explained at high degree, with however some parts needing additional comments or clarification. However, the main concern with this work is the amount of sample patients which is too short to allow authors to draw (unequivocally!) valid conclusions. In line with this, major revisions are needed before publication.
Specific Comments:
1) The authors should revise the paper carefully for English language style, grammar and spelling, and make appropriate corrections and changes.
2) In the Introduction section, authors claimed that there is only one study focused on the role of C1orf74 gene in cancer, why is that? As their research paper is based on this assumption, the author should leave some comments concerning this on the Introduction. I feel that, at some point, the reader needs more information to fully get a picture on the discussed topic..
3) Details concerning patients’ samples are missing (age, race, other diseases, blood type…). The sample has to be characterized.
4) The main concern with this work is the amount of sample patients which is too short to allow authors to draw unequivocally valid conclusions. Do authors have the possibility to increase sample patients? To correlate individual/specific characteristics of patients that may influence and/or contribute for higher expression of C1orf74 gene? Do they search for this?
5) Image resolution of Figure 1 should be improved, as well as statistic analysis. Also, graphs in 1C, D and E have huge deviations, why? Some comment, elucidation should be added…
6) It becomes not very clear if authors investigate the correlation of higher expression of C1orf74 gene and p53 and RB levels, how are they affected? And is this variable for different patients (the ones bearing metastases versus the patients with primary tumor?…).
Comments on the Quality of English Language
The authors should revise the paper carefully for English language style, grammar and spelling, and make appropriate corrections and changes.
Author Response
Author's Reply to the Review Report (Reviewer 2)
The subject of the manuscript is actual and relevant, and this is a good piece of work. The paper is clearly written providing the necessary information (some shortcomings, see below). The observed results are discussed and explained at high degree, with however some parts needing additional comments or clarification. However, the main concern with this work is the amount of sample patients which is too short to allow authors to draw (unequivocally!) valid conclusions.
In line with this, major revisions are needed before publication.
Specific Comments
1) The authors should revise the paper carefully for English language style, grammar and spelling, and make appropriate corrections and changes.
We have carefully revised the paper for English language and made the necessary corrections and changes as suggested by the Reviewer.
2) In the Introduction section, authors claimed that there is only one study focused on the role of C1orf74 gene in cancer, why is that? As their research paper is based on this assumption, the author should leave some comments concerning this on the Introduction. I feel that, at some point, the reader needs more information to fully get a picture on the discussed topic.
We discovered the C1orf74 gene while performing an unbiased search for expression of genes associated with poor prognosis of cervical cancer. In paragraph 3 of Introduction, we have indicated the findings of previous NGS studies and also the rationale for pursuing additional work on this gene in our patient cohort. We have removed the sentence, “This is the only study discussing the role of C1orf74 gene in cancer”. In the introduction and discussion, we have also highlighted the previously studied role of C1orf74 in lung cancer in regulating EGFR/AKT/mTORC1 and how this pathway plays a role even in cervical cancer carcinogenesis through the actions of HPV oncoproteins.
3) Details concerning patients’ samples are missing (age, race, other diseases, blood type…). The sample has to be characterized.
The patient information was reanalysed by a pathologist and additional details are now included in Table 1. We have also mentioned this in the manuscript - “All the tumor samples were characterized by a certified pathologist” in line 158.
4) The main concern with this work is the amount of sample patients which is too short to allow authors to draw unequivocally valid conclusions. Do authors have the possibility to increase sample patients? To correlate individual/specific characteristics of patients that may influence and/or contribute for higher expression of C1orf74 gene? Do they search for this?
We included patient samples in our study to validate the RNA-seq data generated from TCGA analyses. Our analysis fully validated the in silico findings suggesting no significant differences in the expression of C1orf74 among different stages of cervical cancer. We are unable to analyse differences between HPV-positivity and -negativity among our patient cohort as we do not have any HPV-negative cervical cancer patients till date. Unfortunately, we are unable to increase the sample size for this analysis due to several practical constraints, including limited access to additional patients’ samples within a reasonable period of time. In any case, our findings indicated statistically significant differences in the expression of C1orf74 in tumors relative to normal tissues.
5) Image resolution of Figure 1 should be improved, as well as statistic analysis. Also, graphs in 1C, D and E have huge deviations, why? Some comment, elucidation should be added…
We have added a high-resolution image for figure 1. Statistical analysis is available in the supplementary table. The graphs in 1C, D, and E are obtained from public databases like UALCAN and OncoDB. Unfortunately, we do not have control over the deviations in these graphs as they are generated based on the data available at these external resources.
We have discussed that C1orf74 expression in the HPV-positive cell line Ca Ski, which has multiple copies of HPV genomes integrated, is significantly higher compared to HPV-positive cell lines such as HeLa and SiHa where only 1 or 2 copies of the HPV genome are integrated. We have alluded to this point in the last line of paragraph 2 in the Discussion and added an additional comment – “Additionally, there are huge deviations in the expression levels of C1orf74 noted in the in-silico analysis and also in our patient cohort. This is similar to what we observed in HPV-positive cell lines where number of copies of HPV and the location of HPV integration might influence C1orf74 expression. It would be interesting to correlate C1orf74 expression with the HPV integration state and loci in these patients as observed in the aforementioned HPV positive cell lines.”
6) It becomes not very clear if authors investigate the correlation of higher expression of C1orf74 gene and p53 and RB levels, how are they affected? And is this variable for different patients (the ones bearing metastases versus the patients with primary tumor?…).
We agree that this could be confusing especially since we are not analysing the functional role of C1orf74 and its downstream effects. We have removed the sentences speculating the correlation of C1orf74 overexpression with p53 and RB levels from paragraph 6 of the Discussion.
Reviewer 3 Report
Comments and Suggestions for Authors
Parida et al describe an analysis of the C1orf74 gene being a putative biomarker in cervical cancer. The main part of the manuscript is funded by extracting expression and patient data from commonly available databases. On top of this, some expression experiments are done by the authors in cancer cell lines and tumor samples.
Major concerns.
1. In Figure 2 the authors appear to present their expression data relative to a control that represents normal fibroblasts. They use this comparison to state throughout the manuscript that C1orf74 is over-expressed in tumors. This is over-concluded and not sound from a scientific perspective. The authors need to present their expression data using a control/controls much better matching the actual addressed samples and cell lines in terms of cell origin.
2. In Figure 3 differences in DNA methylation are presented for the C1orf74 gene between normal and tumor samples and the identified difference is used as an argument that such DNA methylation changes are functionally associated with the differences in C1orf74 expression. However, looking at the actual differences in DNA methylation in Figure 3 shows beta-value differences in order 0.02! This is very unlikely that this is of major impact. At least, the authors need to argue better that such a small change can have the stated impact. E.g. references describing other genes in cancer that similar DNA methylation differences have a major impact on expression. Concerning this, discussion lines 343-354 appear to be a major overinterpretation of the actual data as well as the use of the wording ‘hypomethylation’ to describe their observations. Throughout the entire manuscript, the observations concerning the very small changes in DNA methylation and the description of these data need extensive revision.
3.
Minor concerns.
1. Table 1. Please write the precise meaning of the abbreviation ‘NA’ used here.
2. Line 77 in the introduction. Here is a need to update common knowledge. A reference to only microarray analyses is outdated. Please acknowledge some NGS-based analyses already here.
3. Line 117. TCGA abbreviation was placed wrongly.
Comments on the Quality of English LanguageGenerally speaking, the English writing is acceptable. However, some typos, misplacement of abbreviations, and incorrect sentence structures are present, and the manuscript will improve with further text editing.
Author Response
Author's Reply to the Review Report (Reviewer 3)
Parida et al describe an analysis of the C1orf74 gene being a putative biomarker in cervical cancer. The main part of the manuscript is funded by extracting expression and patient data from commonly available databases. On top of this, some expression experiments are done by the authors in cancer cell lines and tumor samples.
Major concerns.
- In Figure 2 the authors appear to present their expression data relative to a control that represents normal fibroblasts. They use this comparison to state throughout the manuscript that C1orf74 is over-expressed in tumors. This is over-concluded and not sound from a scientific perspective. The authors need to present their expression data using a control/controls much better matching the actual addressed samples and cell lines in terms of cell origin.
We are sorry for the confusion. Our qPCR data was normalized to both an internal control followed by expression data from paired normal blood sample. We acknowledge your concern, and in response, we have included the data analysis as a excel file titled “Supplementary File-QPCR-Data” supporting our calculations. We have also changed the figure to display gene expression data from matched blood samples from patients in Figure 2B.
- In Figure 3 differences in DNA methylation are presented for theC1orf74 gene between normal and tumor samples and the identified difference is used as an argument that such DNA methylation changes are functionally associated with the differences in C1orf74 expression.
However, looking at the actual differences in DNA methylation in Figure 3 shows beta-value differences in order 0.02! This is very unlikely that this is of major impact. At least, the authors need to argue better that such a small change can have the stated impact. E.g. references describing other genes in cancer that similar DNA methylation differences have a major impact on expression. Concerning this, discussion lines 343-354 appear to be a major overinterpretation of the actual data as well as the use of the wording ‘hypomethylation’ to describe their observations. Throughout the entire manuscript, the observations concerning the very small changes in DNA methylation and the description of these data need extensive revision.
We appreciate the feedback and agree with the suggestion regarding the difference in the average promoter methylation levels in Figure 3. In response, we have revised the Discussion to better acknowledge the limited impact of the observed beta-value differences of 0.02. We also removed the term 'hypomethylation' to avoid overinterpretation of the data. Additionally, we have added the sentence – “Further functional analysis of C1orf74 will help understand if the differences in promoter methylation levels indeed contribute to disease progression”.
Minor concerns.
- Table 1. Please write the precise meaning of the abbreviation ‘NA’ used here.
We have now spelled out NA as “not available.”
- Line 77 in the introduction. Here is a need to update common knowledge. A reference to only microarray analyses is outdated. Please acknowledge some NGS-based analyses already here.
We have added few NGS based analyses as suggested in the Introduction (lines 91 - 96).
- Line 117. TCGA abbreviation was placed wrongly.
We have placed the abbreviation correctly at Line 133.
Round 2
Reviewer 2 Report
Comments and Suggestions for Authors
The authors have made a strong effort to meet the reviewer´s requirements and answered all raised questions and issues. I do recommend the acceptance of the manuscript.
Comments on the Quality of English LanguageThe authors have clearly made an effort to improve the quality of English language, and based on this, the manuscript has been considerable improved.
Reviewer 3 Report
Comments and Suggestions for Authors
All the requests are addressed in the revision